# Parsimonious Predictions for Strategyproof Scheduling

**Richard Cole**     **Anupam Gupta**     **Pranav Jangir**

Department of Computer Science
New York University
New York, NY 10012.

## Abstract

We consider the problem of scheduling $m$ jobs on $n$ unrelated strategic machines to minimize the maximum load of any machine. As the machines are strategic they may misreport processing times to minimize their own load. The pioneering work of Nisan and Ronen gave an $n$-approximate deterministic strategyproof mechanism for this setting, and this was recently shown to be best possible by the breakthrough results of Christodoulou et al. This large approxation guarantee begs the question: how can we avoid these large worst-case results. In this work, we use the powerful framework of algorithms with (machine-learned) predictions to bypass these strong impossibility results. We show how we can predict $O(m + n)$ values to obtain a deterministic strategyproof algorithm whose makespan is within a constant factor of the optimal makespan when the predictions are correct, and $O(n)$ times the optimum no matter how poor the predictions are.

## 1 Introduction

We consider the NP-hard problem of scheduling jobs on unrelated machines to minimize the makespan, i.e., the load of the most loaded machine, in the setting where the machines are agents which can behave strategically. In this game-theoretic setting, introduced by Nisan and Ronen (2001), only the machines know the processing times for the jobs. Since they suffer a cost for processing jobs (i.e., the processing time/size of the job), they have an incentive to misreport sizes (i.e., to overstate or understate them) to the mechanism in order to get a more desirable bundle of jobs. Our goal is to design a mechanism that achieves (a) a good approximation ratio, so that we always find a schedule with makespan comparable to that of the optimal solution on the true processing sizes, and (b) strategyproofness, so that no machine has any incentive to misstate their processing times.

In the non-strategic setting of classical approximation algorithms, the unrelated machines scheduling problem can be approximated to a factor of $2$ (due to works of Lenstra et al. (1990) and Shmoys and Tardos (1993)), and it is NP-hard to approximate it better than a factor of $3/2$, leaving a small gap between the constants in the upper and lower bounds. But when the machines can behave strategically, it is now known that the optimal approximation factor of any *strategyproof* mechanism is not a constant, but exactly $n$. The paper of Nisan and Ronen (2001) gave a mechanism achieving the $n$-approximation, and now a matching lower bound of $n$ for any strategyproof mechanism is known due to Christodoulou et al. (2022, 2023). This leaves us with a large gap between the results that are achievable in the non-strategic and strategic settings in the worst-case.

Since worst-case instances may be pathological and the analysis pessimistic, we can ask: *how can we give mechanisms that perform better in non-worst-case settings?* An important approach of *algorithms with predictions* or *learning-augmented framework*, has gained momentum, giving a balance between the robustness guarantees of the worst-case model, and better results when given

39th Conference on Neural Information Processing Systems (NeurIPS 2025).

machine-learned information. Concretely, the model asks: can we perform better if we have some (machine-learned) untrusted predictions about the instance? We seek the following two outcomes:

1. *Robustness:* this requires that the mechanism's performance is not much worse than the performance without predictions, regardless of whether the predictions are correct or not.
2. *Consistency:* this asks that if the predictions are indeed correct (i.e., the predictions indeed conform to the parameters/features of the true instance they are supposed to be predicting), then the mechanism's performance is close to that achievable if it truly believed the predictions. We may also ask for an error-tolerant version of consistency, where the algorithm's performance relates to the "error" in the predictions.

This line of work was initiated in the setting of online algorithms by Mahdian et al. (2012), Purohit et al. (2018) and Lykouris and Vassilvitskii (2021) (see also the survey article by Mitzenmacher and Vassilvitskii (2022)); it was used in strategic settings by Agrawal et al. (2024), Xu and Lu (2022) and Gkatzelis et al. (2022). (See the annotated bibliography compiled by Balkanski et al. (2024)).

The unrelated machine scheduling problem was studied in this learning-augmented setting by Xu and Lu (2022) and Agrawal et al. (2024), who observed that following the predictions blindly gives an excellent consistency of 1, but has arbitrarily poor robustness if the predictions are faulty; to remedy this, they gave algorithms with bounded robustness and consistency. The current best result is due to Balkanski et al. (2023), who gave the SCALED GREEDY algorithm that is $(1 + c)n$-robust and $(4 + 2/c)$-consistent, for any $c > 0$. Their algorithm uses the entire matrix $\widehat{p} \in \mathbb{R}_+^{n \times m}$ of processing times, where $\widehat{p}_{ij}$ is the prediction for the true processing time $p_{ij}$ of the job $j$ on machine $i$.

Christodoulou et al. (2024) explore whether we can use a smaller set of predictions, namely *output predictions*, where the mechanism is merely given the predicted machine for each job. They give an algorithm that uses just these $m$ many predictions, and that is $O(1)$-robust and $(n^2)$-consistent. Their algorithm—and indeed, all the above algorithms in this domain—compute machine "biases". They use a greedy-like algorithm using the reported processing times weighted by these biases to assign the jobs. They also show that, given just assignment predictions, no such weighted VGC mechanism can obtain much better trade-offs.

In light of these results, we ask: can we get a different set of linearly many predictions, i.e., $O(n+m)$ quantities, and achieve the same guarantees?

## 1.1 Our Results

Our main result answers this question in the affirmative, and gives nearly optimal robustness/consistency tradeoffs with only a linear number of predictions.

**Theorem 1.1** (Informal Main Theorem). *There exists a strategyproof mechanism that takes predictions for the value of the optimal makespan, a "bias" for each machine, and a "good" machine for each job, and produces an assignment which is $(1 + c)n$-robust, and $(2 + \frac{1}{c-1})$-consistent for any choice of $c > 1$.*

This achieves essentially the same guarantees as the result of Balkanski et al. (2023) (our results replace a $1/c$ term in the consistency guarantee by $1/(c-1)$), while using far fewer predictions—only linearly many instead of quadratically many. And using these few extra predictions (corresponding to the "biases" for each machines) to avoid the lower bounds of Christodoulou et al. (2024).

Our approach is based on an essentially simple idea: instead of trying to predict the optimal machines for each job, we predict the machines which would be output by an approximation algorithm. This algorithm follows the "relax-and-round" approach: it solves a linear programming (LP) relaxation for the unrelated machine scheduling problem, and then converts the fractional solution into an integral one. Crucially, the assignments given by this rounding algorithm are supported by linear programming duality. We use this in two ways: firstly, we use the machine variables of the dual program as the machine biases. Secondly, we use the structure of the dual solutions (e.g., complementary slackness) along with the output of the rounding process to prove robustness guaranteees.

While this approach leads to an $O(n^2)$-robust algorithm, the dual structure of the classical LP relaxation is not rigid enough. The next idea is to write a new LP relaxation for unrelated machine scheduling problem, which allows us to ensure that the duals are bounded, which gives us more

control over the robustness. The final ingredient is showing a good approximation guarantee for this new relaxation.

Finally, like Balkanski et al. (2023), we can extend our work to an error-tolerant version. In this setting, we are given a parameter $\eta > 1$, and predictions are considered "$\eta$-good" if they correspond to some processing times that are within a factor of $\eta$ of the true processing times. The goal now is to be consistent when the predictions are "$\eta$-good", otherwise the algorithm is required to be robust. The formal definitions and details of the following result appear in Appendix B.

**Theorem 1.2** (Informal Error-Tolerant Theorem). *There exists a strategyproof mechanism that takes the above predictions for the value of the optimal makespan, the optimal dual solution and the predicted rounded solution, and produces an assignment which is $(1 + c)\,\eta^2\,n$-robust (when the predictions are not $\eta$-good), and $\widehat{\eta}^2\gamma(2 + \frac{1}{c-1})$-consistent (when they are $\widehat{\eta}$-good for some $\widehat{\eta} \le \eta$) for any choice of $c > 1$.*

**Comparing the Prediction Models.**   An advantage of parsimonious predictions is that they require only linear space to manipulate and store, compared to the quadratic storage used in the works ofXu and Lu (2022), Agrawal et al. (2024), Balkanski et al. (2023). It is an open problem to prove that learning these smaller number predictions from data has a smaller sample complexity. (We briefly discuss the learnability of the processing times in a PAC-like model in Appendix A.) That said, we can use the predictions given to Balkanski et al. (2023) (i.e., predictions for all the processing times) to obtain our results by first computing the predictions that our mechanism expects (which are all functions of the processing times), and using these $1 + n + m$ values for our mechanism.

## 1.2 Other Related Work

Strategyproof mechanisms are increasingly important in open and decentralized systems, since the resources belong to different agents who may act in their own interest and may not cooperate unless they are given the right incentives. In such settings, it is often important to ensure notions of non-manipulability and fairness, either for reasons of social welfare, or because of legal requirements. Strategyproof mechanisms can help ensure that no single agent can manipulate the outcome to their advantage. Specifically for settings of scheduling jobs to machines, note that machines may misreport processing times to reduce their workload, which could lead to poor scheduling decisions and delays for jobs. The first work for the unrelated machine scheduling problem in the strategyproof setting was due to Nisan and Ronen (2001). The lower bounds were given by Nisan and Ronen (2001), Christodoulou et al. (2009), Koutsoupias and Vidali (2013), Giannakopoulos et al. (2021), Dobzinski and Shaulker (2020), Ashlagi et al. (2012), Christodoulou et al. (2007), culminating in a matching lower bound of $n$ in the recent works of Christodoulou et al. (2022, 2023).

The work of Christodoulou et al. (2024) considers the *output predictions* model, where the mechanism is merely given a predicted output for each request: this corresponds to a predicted machine for each job in our setting. They give an algorithm that is $(\beta + 1)$-robust and $(n^2/\beta)$-consistent for any fixed $\beta \in [1, n]$; it assigns a bias of $r_{ij} = 1$ for the predicted machine $i$, and $r_{i'j} = n/\beta$ for every other machine $i'$, and chooses the weighted minimizer of $r_{ij}p_{ij}$. They also show that, given just assignment predictions, no weighted VGC mechanism can give a significantly better trade-off than $\beta$ vs. $n^2/\beta$. In contrast, our work shows that if we are also given a prediction for OPT and the dual variables (one for each machine), we can achieve much stronger results.

The use of linear programs and duality as predictions is widespread: they have been used, e.g., the works of Lavastida et al. (2021), Lattanzi et al. (2020), Li and Xian (2021) uses linear/convex programs and their duals to perform load balancing in the online setting. Moreover, linear programming duals are used in the work of Dinitz et al. (2021); it uses learned duals as a "warm start" to speed up algorithms for finding maximum weight bipartite matchings; these ideas are further explored by Davies et al. (2023), Polak and Zub (2024), Sakaue and Oki (2022). However, our use in mechanism design, and in particular, our approach of controlling the variation in the duals to make the mechanism more "robust", seems to not have been used before.

## 2   Definitions

An instance $\mathcal{I}$ of the unrelated machine scheduling problem consists of (i) a set $J$ of $m$ jobs, (ii) a collection $M$ of $n$ machines, and (iii) a processing time $p_{ij}$ for each job $j \in J$ and machine $i \in M$.

Let $p$ denote the vector of all processing times. Given an instance $\mathcal{I}$, an assignment of jobs to machines is a vector $\boldsymbol{x} \in \{0, 1\}^{n \times m}$, where $x_{ij} = 1$ if and only if job $j$ is assigned to machine $i$. Such an assignment $\boldsymbol{x}$ satisfies $\sum_i x_{ij} = 1$, and induces a *load* of $\sum_j p_{ij} x_{ij}$ on machine $i \in M$.

- The *makespan* of assignment $\boldsymbol{x}$ is $\mathsf{MS}(\boldsymbol{p}, \boldsymbol{x}) := \max_{i \in M} \sum_j p_{ij} x_{ij}$.
- Define the optimal (integer) makespan to be $\mathsf{OPT}(\boldsymbol{p}) := \min_{\text{assignments } \boldsymbol{x}} \mathsf{MS}(\boldsymbol{p}, \boldsymbol{x})$, where the minimum is taken over all possible assignments $\boldsymbol{x}$ of jobs to machines, and the optimal (integer) assignment be denoted by $\boldsymbol{x}^\star = \boldsymbol{x}^\star(\boldsymbol{p})$.

Given an assignment $\boldsymbol{x}$, we can also associate an allocation function $\varphi : J \to M$ with it, where $\varphi(j) = i \iff x_{ij} = 1$. We frequently move between allocation functions $\varphi$ and assignments $\boldsymbol{x}$, as needed. (In some cases, we consider $\boldsymbol{x} \in \{0, 1\}^{n \times m}$ vectors where $\sum_i x_{ij} \geq 1$, but we can always reduce some of the $x_{ij}$ values to zero and get $\sum_i x_{ij} = 1$, without increasing the machine loads and the makespans.)

## 2.1 Strategyproofness

Consider an instance of the unrelated machine scheduling problem, where the machines report their processing times for all the jobs. Let $\boldsymbol{p} \in \mathbb{R}_+^{n \times m}$ be the true processing times for all machine-job pairs. (As is standard, we use $\boldsymbol{p}_i$ to denote the $i^{th}$ row of matrix $\boldsymbol{p}$, and $\boldsymbol{p}_{-i}$ to denote the rest of the rows.) Now suppose some machine $i \in M$ reports $\boldsymbol{p}'_i \in \mathbb{R}_+^m$ as its vector of processing times (where $\boldsymbol{p}'_i$ may or may not equal $\boldsymbol{p}_i$), and the other machines truthfully report $\boldsymbol{p}_{-i} \in \mathbb{R}_+^m$:

- let $\boldsymbol{x}(\boldsymbol{p}'_i, \boldsymbol{p}_{-i}) \in \{0, 1\}^{n \times m}$ be the assignment of jobs to machines, and
- let $\pi(\boldsymbol{p}'_i, \boldsymbol{p}_{-i}) \in \mathbb{R}_+^{n \times m}$ be the payments made by the mechanism to machines.

Define the *utility* of machine $i$ to be

$$U_i(\boldsymbol{p}'_i, \boldsymbol{p}_{-i}) := \sum_j \pi_{i,j}(\boldsymbol{p}'_i, \boldsymbol{p}_{-i}) - \sum_j p_{ij} \boldsymbol{x}_{ij}(\boldsymbol{p}'_i, \boldsymbol{p}_{-i}). \tag{1}$$

Observe that while both the map $\boldsymbol{x}$ and the payments $\pi$ are functions of the reports $(\boldsymbol{p}'_i, \boldsymbol{p}_{-i})$, the second term in the definition of utility uses the true processing times.

**Definition 2.1** (Strategyproofness). An allocation function $\varphi$ is *strategyproof* if there exists payments $\pi$ such that for any machine $i$ and for any misreport of the processing times vector $\boldsymbol{p}'_i$, we have $U_i(\boldsymbol{p}_i, \boldsymbol{p}_{-i}) \geq U_i(\boldsymbol{p}'_i, \boldsymbol{p}_{-i})$.

**Definition 2.2** (Monotonicity). Let $\boldsymbol{x} = \boldsymbol{x}(\boldsymbol{p}_i, \boldsymbol{p}_{-i})$ and $\boldsymbol{x}' = (\boldsymbol{p}'_i, \boldsymbol{p}_{-i})$ denote the allocations of the mechanism when machine $i$ reports $\boldsymbol{p}_i$ and $\boldsymbol{p}'_i$ respectively. An allocation mechanism is *monotone* if for every machine $i$ and for every two reports of the machine $\boldsymbol{p}_i$ and $\boldsymbol{p}'_i$, the associated allocations satisfy $\sum_{j \in J}(x_{ij} - x'_{ij})(p_{ij} - p'_{ij}) \leq 0$. Moreover, call the mechanism *item-wise monotone* if the allocations satisfy $(x_{ij} - x'_{ij})(p_{ij} - p'_{ij}) \leq 0 \qquad \forall j \in J$.

**Theorem 2.3** (Nisan and Ronen (2001), Saks and Yu (2005)). *For an allocation algorithm, there exists a payment function $\pi$ such that $(\varphi, \pi)$ is strategyproof if and only if the allocation algorithm satisfies the monotonicity property.*

Since any item-wise monotone mechanism is also monotone, the following corollary holds:

**Corollary 2.4.** *An allocation algorithm is strategyproof if it satisfies the item-wise monotonicity property. In other words, if for every machine $i$, suppose $\boldsymbol{x} = \boldsymbol{x}(\boldsymbol{p}_i, \boldsymbol{p}_{-i})$ and $\boldsymbol{x}' = (\boldsymbol{p}'_i, \boldsymbol{p}_{-i})$, and suppose these allocations satisfy $(x_{ij} - x'_{ij})(p_{ij} - p'_{ij}) \leq 0$ for every job $j$. Then the allocation algorithm is monotone, and hence there exists a payment function that gives strategyproofness.*

## 3 A New Linear Program Relaxation

We now give a new linear programming relaxation for the scheduling problem, such that the dual variables corresponding to the machines lie in some bounded range. It is this control on the dual variables allows us to show strong robustness guarantees without losing much on the consistency.

The standard linear programming relaxation for the unrelated machine scheduling problem takes an instance $\mathcal{I}$ and a target makespan $T$, and defines the set of *permissible edges* $E(T, \boldsymbol{p}) := \{(i, j) \mid$

$p_{ij} \leq T\}$ by removing all the edges in $M \times J$ which correspond to processing times larger than the target value $T$. The *feasibility* linear program $P(T, \boldsymbol{p})$ is then defined as follows:

$$\sum_{i:(i,j)\in E(T,\boldsymbol{p})} x_{ij} \geq 1$$
$$\sum_{j:(i,j)\in E(T,\boldsymbol{p})} p_{ij}\, x_{ij} \leq T \qquad \forall \text{ machines } i \in M \qquad (2)$$
$$x \geq 0.$$

Again, the first constraints allow the fractional assignment to allocate more than 1 unit of job $j$, but any such solution can be transformed into one that satisfies these constraints with equality. Using inequalities simplifies the linear programming dual $D(T, \boldsymbol{p})$:

$$\max \sum_j \alpha_j - T \sum_i \beta_i \qquad (D(T,\boldsymbol{p}))$$
$$\alpha_j - \beta_i\, p_{ij} \leq 0 \qquad \forall (i,j) \in E(T,\boldsymbol{p})$$
$$\alpha, \beta \geq 0.$$

Our new linear programming relaxation for the unrelated machine scheduling problem takes an instance $\mathcal{I}$ and a target makespan $T$, and defines the set of *permissible edges* $E(T,\boldsymbol{p}) := \{(i,j) \mid p_{ij} \leq T\}$ by removing all the edges in $M \times J$ which correspond to processing times larger than the target value $T$. Then for any a scalar $c \geq 1$, the linear program relaxation $P_c(T, \boldsymbol{p})$ is the following:

$$\min \quad Z - \nicefrac{1}{cn} \sum_i Y_i \qquad (P_c(T,\boldsymbol{p}))$$
$$\sum_{i:(i,j)\in E(T,\boldsymbol{p})} x_{ij} \geq 1 \qquad \forall \text{ jobs } j \in J$$
$$\sum_{j:(i,j)\in E(T,\boldsymbol{p})} p_{ij}\, x_{ij} - Z + Y_i \leq T \qquad \forall \text{ machines } i \in M$$
$$\boldsymbol{x}, \boldsymbol{Y}, Z \geq 0.$$

In other words, the primal constraint for machine $i \in M$ says:

$$\sum_{j:(i,j)\in E(T,\boldsymbol{p})} p_{ij}\, x_{ij} + Y_i \leq T + Z. \qquad (3)$$

The first constraints assigns each job to some machine, and is a natural one. But the second constraint is mysterious due to the variables $Y_i$ and $Z$: to demystify this, let us write its dual program $D_c(T, \boldsymbol{p})$:

$$\max \sum_{j \in J} \alpha_j - T \sum_{i \in M} \beta_i \qquad (D_c(T,\boldsymbol{p}))$$
$$\alpha_j - \beta_i\, p_{ij} \leq 0 \qquad \forall (i,j) \in E(T,\boldsymbol{p})$$
$$\sum_{i \in M} \beta_i \leq 1 \qquad (4)$$
$$\beta_i \geq \nicefrac{1}{cn} \qquad \forall i \in M \qquad (5)$$
$$\alpha, \beta \geq 0.$$

Note that the dual program has variables $\alpha_j$ for each job $j \in J$, and variables $\beta_i$ for each machine $i \in M$; the latter should be thought of as machine "weights". Observe that this LP is very similar to the dual for the natural relaxation, where constraints (4) and (5) are the new constraints added to control the range of the $\beta_i$ values. Moreover, complementary slackness says that $x_{ij}(\alpha_j - \beta_i\, p_{ij}) = 0$.

**Proposition 3.1.** If $T \geq OPT(\boldsymbol{p})$, then both the primal and dual linear programs above are feasible.

*Proof.* Indeed, consider the assignment $\boldsymbol{x}^\star(\boldsymbol{p})$ that achieves the optimal makespan for input $\boldsymbol{p}$; it is a feasible solution to the primal program. Moreover, the all-zero solution is feasible for the dual program. $\square$

## 3.1 Properties of the New Programs

The optimal solution for the new dual program $D_c(T, \boldsymbol{p})$ has a value no larger than that of the standard dual, since it is a maximization problem with more constraints. Of course, the added constraints could conceivably make the dual infeasible, but we now show that this is not possible.

**Lemma 3.2** (LPs are Bounded). *For any $c > 1$ and any $T \geq OPT(\boldsymbol{p})$, the primal $P_c(T, \boldsymbol{p})$ is feasible, and any (fractional) solution $(\boldsymbol{x}, \boldsymbol{Y}, Z)$ to it has value at least $Z(1 - \nicefrac{1}{c}) - \nicefrac{T}{c} \geq -\nicefrac{T}{c}$. Moreover, its dual $D_c(T, \boldsymbol{p})$ is feasible and bounded as well.*

*Proof of Lemma 3.2.* The original primal $P(T, \boldsymbol{p})$ is feasible because of Proposition 3.1; moreover, any solution for the original primal LP is also feasible for the new primal $P_c(T, \boldsymbol{p})$ by just setting $Z = Y_i = 0$. To show the bound on the primal objective value, consider the constraint (3). The non-negativity of the sum $\sum p_{ij} x_{ij}$ implies that $Y_i \leq T + Z$. Consequently, the primal objective value is

$$Z - \tfrac{1}{cn} \sum_i Y_i \geq Z - \tfrac{n(T+Z)}{cn} = Z\left(1 - \tfrac{1}{c}\right) - \tfrac{T}{c} \geq -\tfrac{T}{c},$$

where we used the non-negativity of the $Z$ variable in the final step. Since the primal $P_c(T, \boldsymbol{p})$ is feasible and bounded from below, we can use strong duality to infer that the dual $D_c(T, \boldsymbol{p})$ is also feasible and bounded, which completes the proof. $\qquad\square$

While the objective function for the primal $P_c(T, \boldsymbol{p})$ is not meaningful *per se*, we can also show that the optimal fractional solution has small makespan.

**Lemma 3.3** (Fractional Makespan). *For $c > 1$ and target $T \geq OPT(\boldsymbol{p})$, an optimal fractional solution $(\widetilde{\boldsymbol{x}}, \widetilde{\boldsymbol{Y}}, \widetilde{Z})$ for the linear program $P_c(T, \boldsymbol{p})$ has (fractional) makespan*

$$\max_i \sum_{j:(i,j)\in E(T,\boldsymbol{p})} p_{ij}\widetilde{x}_{ij} \;\; \leq \;\; T \cdot \tfrac{c}{c-1}.$$

*Proof.* Since any solution to the classical primal is also a solution to the new primal (by setting $Z = Y_i = 0$), the optimal solution to the primal $P_c(T, \boldsymbol{p})$ has objective value at most zero. Moreover, Lemma 3.2 implies that the optimal solution to $P_c(T, \boldsymbol{p})$ has objective value at least $\widetilde{Z}(1 - 1/c) - T/c$. Combining these facts,

$$\widetilde{Z}\left(1 - \tfrac{1}{c}\right) \leq \tfrac{T}{c} \implies \widetilde{Z} \leq \tfrac{T}{c-1}.$$

Using constraint (3) again, but this time ignoring the non-negative $\widetilde{Y}_i$ terms, we get that

$$\sum_{j:(i,j)\in E(T,\boldsymbol{p})} p_{ij}\widetilde{x}_{ij} \leq T + \widetilde{Z} \leq T\left(1 + \tfrac{1}{c-1}\right)$$

for each machine $i \in M$, as claimed. $\qquad\square$

In other words, any optimal fractional solution $\widetilde{\boldsymbol{x}}$ for $P_c(T, \boldsymbol{p})$ is also a solution to the following relaxed version of the original linear program $P(T, \boldsymbol{p})$:

$$\sum_{i:(i,j)\in E(T,\boldsymbol{p})} x_{ij} \geq 1 \qquad \forall \text{ jobs } j \in J \qquad\qquad (Q_\lambda(T,\boldsymbol{p}))$$

$$\sum_{j:(i,j)\in E(T,\boldsymbol{p})} p_{ij} x_{ij} \leq \lambda T \qquad \forall \text{ machines } i \in M \qquad\qquad (6)$$

$$x \geq 0,$$

as long as we set $\lambda = \tfrac{c}{c-1}$. (Note that when the relaxation parameter $\lambda = 1$, then $Q_1(T, \boldsymbol{p}) = P(T, \boldsymbol{p})$, i.e., we get back the original primal linear program.)

## 3.2 Rounding Algorithm

Assume that $T \geq OPT(\boldsymbol{p})$. In this case, the rounding algorithms of Lenstra et al. (1990) and Shmoys and Tardos (1993) take a fractional solution $\widetilde{\boldsymbol{x}} \geq 0$ satisfying the relaxed LP $Q_\lambda(T, \boldsymbol{p})$ and output an integer assignment $\boldsymbol{x}^\dagger \in \{0,1\}^{n \times m}$ with the following properties:

1. $\sum_i x_{ij}^\dagger = 1$ so that each job is assigned to exactly one machine,
2. each job $j$ is assigned to some machine $i$ to which it was originally fractionally assigned in $\widetilde{\boldsymbol{x}}$ (i.e., $x_{ij}^\dagger = 1 \implies \widetilde{x}_{ij} > 0$), and moreover
3. $\boldsymbol{x}^\dagger$ is also a solution to $Q_{\lambda'}(T, \boldsymbol{p})$ but now with $\lambda' = \lambda + 1$.

In other words, this map satisfies

$$\mathsf{MS}(\boldsymbol{p}, \boldsymbol{x}^\dagger) \leq \max_i \sum_{j:(i,j)\in E(T,\boldsymbol{p})} p_{ij} x_{ij}^\dagger \leq (\lambda + 1) \cdot T = (2 + 1/(c-1)) \cdot T. \qquad (7)$$

This analysis can be tightened a bit to make the additive loss depend on the largest processing time of any job in the optimal solution, but we stick to the analytical form (7) for simplicity.

## 4 The Mechanism and the Predictions

Our mechanism takes the following predictions: (i) a predicted threshold $\widehat{T}$, (ii) predicted dual variables $\widehat{\beta}_i \in [1/cn, 1]$ for each machine $i \in M$, and (iii) a predicted assignment $\widehat{\varphi} : J \to M$. In the ideal solution, the threshold $\widehat{T}$ should equal $\mathsf{OPT}(\boldsymbol{p})$, the variables $\widehat{\boldsymbol{\beta}}$ should be an optimal solution to the modified dual program $D_c(T, \boldsymbol{p})$, and the predicted assignment $\widehat{\boldsymbol{x}}$ should be the assignment $\boldsymbol{x}^{\dagger}$ arising from optimally solving the modified primal $P_c(T, \boldsymbol{p})$ and then rounding it as in Section 3.2. The predicted assignment $\widehat{\varphi}$ corresponds to a vector $\widehat{\boldsymbol{x}} \in \{0, 1\}^{n \times m}$, and henceforth we will use the two objects interchangeably. Finally, observe that if we are instead given a prediction $\widehat{\boldsymbol{p}}$ of the processing times, we can compute the values above—as described in Section 3—and use them instead.

### 4.1 The Mechanism

To begin, the DUAL-PREDICTOR mechanism checks whether the predicted dual variables satisfy $\sum_i \widehat{\beta}_i \leq 1$ and $\min_i \widehat{\beta}_i \geq 1/cn$. If not, it rejects the predictions and just runs the greedy algorithm, which is $n$-robust and truthful Nisan and Ronen (2001). If the $\widehat{\beta}$ values are in the correct range, the mechanism considers the reported processing times $\boldsymbol{p} \in \mathbb{R}^{n \times m}$ along with the predictions and does the following:

---

**Mechanism 1:** DUAL-PREDICTOR

**Input:** Reported values $p_{ij}$ for all $(i, j) \in M \times J$, predictions $\widehat{\boldsymbol{\beta}} \in \mathbb{R}^n_+$, map $\widehat{\varphi} : J \to M$, and $\widehat{T} \in \mathbb{R}_+$.

**Output:** Assignment $\boldsymbol{x} \in \{0, 1\}^{n \times m}$.

**for** *each job $j$* **do**

    Let $\mathsf{small}(j) := \{i \mid p_{ij} \leq \widehat{T}\}$.

    **if** $\mathsf{small}(j) = \varnothing$ **then**

        $\varphi(j) \leftarrow \arg\min_i p_{ij}$.

    **else**

        $\varphi(j) \leftarrow \arg\min_i \{\widehat{\beta}_i \, p_{ij} \mid i \in \mathsf{small}(j)\}$.         // breaking ties in favor of $\widehat{\varphi}(j)$.

**return** the vector $\boldsymbol{x}$ corresponding to this allocation $\varphi$.

---

In words, the DUAL-PREDICTOR mechanism builds a set of machines $\mathsf{small}(j) := \{j \mid p_{ij} \leq \widehat{T}\}$ on which the job is "small" compared to our prediction for the optimal makespan, based on the reported sizes $\boldsymbol{p}$ and the prediction $\widehat{T}$ for the optimal makespan. If this set is empty (a sure indicator of the predictions being incorrect), it assigns the job to the machine on which it has the smallest reported size. Else it assigns it to a machine in $\mathsf{small}(j)$ that minimizes $\widehat{\beta}_i p_{ij}$, breaking ties in favor of the predicted machine $\widehat{\varphi}(j)$.

A technical observation: our mechanism uses the reported $p_{ij}$ values to define the set $\mathsf{small}(j)$ (which can be thought of as modifying the biases $\beta_i$). This is a departure from previous algorithms, which define the biases $r_{ij}$ based on the predictions and the reported processing times.

## 5 The Analysis

We now analyze the quality of the assignment $\varphi$ returned by the DUAL-PREDICTOR mechanism: our proof proceeds in three natural steps.

We first show $(1 + c)n$-robustness of the mechanism by proving that no matter what the predictions are, the makespan of the assignment produced by the mechanism is at most $(1 + c)n \cdot \mathsf{OPT}(\boldsymbol{p})$.

Next, we show $(2 + 1/(c-1))$-consistency. We say that when the predictions are correct (i.e., when $\widehat{T} \approx \mathsf{OPT}(\boldsymbol{p})$, and the $\widehat{\beta}_i$ values correspond to an optimal solution to the dual $D_c(\widehat{T}, \boldsymbol{p})$, and also the predicted assignment $\widehat{\varphi}$ corresponds to the integer allocation $\boldsymbol{x}^{\dagger}$ obtained taking an optimal solution $\widetilde{\boldsymbol{x}}$ to the primal $P_c(\widehat{T}, \boldsymbol{p})$ and rounding it as in Section 3.2), then our algorithm achieves a makespan of at most $(2 + 1/c-1) \cdot \widehat{T}$; i.e., a nearly optimal solution.

Finally we show that the mechanism is strategyproof, by showing that the mechanism is monotone.

## 5.1 Robustness

For robustness, we show that no matter what the predictions are, the mechanism's assignment $\boldsymbol{x}$ has makespan $\mathsf{MS}(\boldsymbol{p}, \boldsymbol{x})$ that is at most $O(n)$ times the optimal makespan. Let $\boldsymbol{x}^\star$ be the optimal assignment, so that $\mathsf{OPT}(\boldsymbol{p}) = \mathsf{MS}(\boldsymbol{p}, \boldsymbol{x}^\star)$.

**Theorem 5.1.** *Given any predictions,* $\mathsf{MS}(\boldsymbol{p}, \boldsymbol{x}) \le (1 + c)n \cdot \mathsf{MS}(\boldsymbol{p}, \boldsymbol{x}^\star)$.

*Proof.* Let $\varphi$ and $\varphi^\star$ be maps from $J$ to $M$ that correspond to the assignments $\boldsymbol{x}$ and $\boldsymbol{x}^\star$ in the natural way. Define $r_j := \max(1, cn \cdot \widehat{\beta}_{\varphi^\star(j)})$. To begin, we claim that for any job $j$,

$$p_{\varphi(j),j} \le r_j \cdot p_{\varphi^\star(j),j}. \tag{8}$$

Indeed, consider the various cases:

1. If $\mathsf{small}(j) = \varnothing$, then $p_{\varphi(j),j} \le p_{i,j}$ for all machines $i$, and hence also for machine $\varphi^\star(j)$.

2. Else if $|\mathsf{small}(j)| \ge 1$, we consider two subcases.

   (a) If $\varphi^\star(j) \notin \mathsf{small}(j)$, then $\widehat{T} < p_{\varphi^\star(j),j}$. Moreover, we assign job $j$ to some job in $\mathsf{small}(j)$, and therefore $p_{\varphi(j),j} \le \widehat{T}$. Chaining the two implies $p_{\varphi(j),j} < p_{\varphi^\star(j),j}$.
   (b) Else both $\varphi(j)$ and $\varphi^\star(j)$ belong to $\mathsf{small}(j)$. The mechanism's selection criterion ensures that $\widehat{\beta}_{\varphi(j)} \, p_{\varphi(j),j} \le \widehat{\beta}_{\varphi^\star(j)} \, p_{\varphi^\star(j),j}$. Since the $\widehat{\beta}$ values range between $1/cn$ and $1$, the claim immediately follows.

This proves the claim (8). Now consider any machine $i$, and let $J_i := \{j \mid \varphi(j) = i\}$ be the jobs assigned to machine $i$ by the mechanism. The load of machine $i$ is

$$\sum_{j \in J_i} p_{i,j} \overset{(8)}{\le} \sum_{j \in J_i} (1 + cn \cdot \widehat{\beta}_{\varphi^\star(j)}) \cdot p_{\varphi^\star(j),j} = \sum_{i' \in M} (1 + cn \cdot \widehat{\beta}_{i'}) \cdot \sum_{j \in J_i : \varphi^\star(j) = i'} p_{i',j}$$

$$\le \sum_{i' \in M} (1 + cn \cdot \widehat{\beta}_{i'}) \cdot \mathsf{MS}(\boldsymbol{p}, \varphi^\star) = (n + cn) \cdot \mathsf{MS}(\boldsymbol{p}, \boldsymbol{x}^\star),$$

where we use $\sum_i \widehat{\beta}_i \le 1$ in the final step. $\square$

## 5.2 Consistency

We now show the consistency guarantees: when the predictions are correct, the makespan of the mechanism's assignment is at most $(2 + \frac{1}{c-1})$ times the optimal makespan. We consider the following slightly relaxed definition:

**Definition 5.2** ($\gamma$-Correct Predictions). For $\gamma \ge 1$, the predictions $(\widehat{T}, \widehat{\beta}, \widehat{\varphi})$ are $\gamma$-*correct* for processing times $\boldsymbol{p}$ if the following conditions hold:

- $\widehat{T} \in [\mathsf{OPT}(\boldsymbol{p}), \gamma \, \mathsf{OPT}(\boldsymbol{p})]$,
- $\widehat{\beta} = \widetilde{\beta}$ for an optimal solution $(\widetilde{\alpha}, \widetilde{\beta})$ to the dual program $D_c(\widehat{T}, \boldsymbol{p})$, and
- the assignment $\widehat{\boldsymbol{x}}$ corresponding to the prediction $\widehat{\varphi}$ is the assignment $\boldsymbol{x}^\dagger$ obtained by rounding an optimal solution $\widetilde{\boldsymbol{x}}$ to the primal $P_c(\widehat{T}, \boldsymbol{p})$, as in Section 3.2.

When predictions are 1-correct, we refer to them as *correct* predictions. We show that $\gamma$-correct predictions imply a good makespan.

**Lemma 5.3** (Consistency). *If the predictions* $(\widehat{T}, \widehat{\beta}, \widehat{\varphi})$ *are* $\gamma$-*correct for processing times* $\boldsymbol{p}$, *then*

$$\mathsf{MS}(\boldsymbol{p}, \boldsymbol{x}) \le (2 + 1/(c-1)) \cdot \widehat{T} \le (2 + 1/(c-1)) \cdot \gamma \, \mathsf{OPT}(\boldsymbol{p}).$$

*Proof.* Observe that the set $\mathsf{small}(j)$ is exactly the set of permissible edges from $E(\widehat{T}, \boldsymbol{p}) := \{(i, j) \mid p_{ij} \le \widehat{T}\}$ which are incident to job $j$. Since $\widehat{T} \ge \mathsf{OPT}(\boldsymbol{p})$, the set $\mathsf{small}(j)$ is non-empty since it contains the machine that $j$ is assigned to in the optimal integer solution $\boldsymbol{x}^\star$. The mechanism now

assigns job $j$ to the machine $\varphi(j) \in \mathsf{small}(j)$ minimizing $\widehat{\beta}_i \, p_{ij} = \widetilde{\beta}_i \, p_{ij}$. We claim that this machine is in fact $\widehat{\varphi}(j)$. The claim implies that $\boldsymbol{x}$ (the output of the mechanism) equals $\widehat{\boldsymbol{x}}$ (the assignment corresponding to the predicted allocation), which in turn equals $\boldsymbol{x}^\dagger$ (the rounded solution). Hence (7) gives us $\mathsf{MS}(\boldsymbol{p}, \boldsymbol{x}) = \mathsf{MS}(\boldsymbol{p}, \widehat{\boldsymbol{x}}) \leq (2 + 1/(c-1)) \cdot \widehat{T}$, which completes the proof.

We now prove the claim that $\varphi(j) = \widehat{\varphi}(j)$. By the property of the rounding algorithm, we know that $\widehat{x}_{i,j} = 1$ only if $\widetilde{x}_{i,j} > 0$. Moreover, we have $\widetilde{x}_{i,j} > 0$ only for $(i, j) \in E(\widehat{T}, \boldsymbol{p})$ and hence $\widehat{\varphi}(j)$ belongs to $\mathsf{small}(j)$. We now claim that

$$\widehat{\beta}_{\widehat{\varphi}(j),j} \, p_{\widehat{\varphi}(j),j} \leq \widehat{\beta}_{ij} \, p_{ij} \tag{9}$$

for all $i \in \mathsf{small}(j)$. Indeed, observe that $\widetilde{\alpha}_j \leq \widetilde{\beta}_i \, p_{ij}$ for all $(i, j) \in \mathsf{small}(j)$ because of dual feasibility. Moreover, complementary slackness implies that $\widetilde{x}_{ij}(\widetilde{\alpha}_j - \widetilde{\beta}_i \, p_{ij})$ for all $i \in \mathsf{small}(j)$. Since $\widetilde{x}_{\widehat{\varphi}(j),j} > 0$ we have that that $\widetilde{\alpha}_j = \widetilde{\beta}_{\widehat{\varphi}(j)} \, p_{\widehat{\varphi}(j),j}$. Moreover, using that $\widehat{\boldsymbol{\beta}} = \widetilde{\boldsymbol{\beta}}$, we get that $\widehat{\varphi}(j)$ is one of the minimizers of $\widehat{\beta}_i \, p_{ij}$, proving (9). Finally, our tie-breaking strategy ensures that the mechanism assigns the job $j$ to $\widehat{\varphi}(j)$. $\qquad\square$

As an aside, note that while we used the fact that $\widehat{\varphi}$ is obtained from the rounding algorithm of Section 3.2, any $\widehat{\varphi}$ satisfying the properties above (in particular, that $\widetilde{x}_{\widehat{\varphi}(j),j} > 0$), and having low load, could be used instead.

## 5.3 Strategyproofness

To show that the the mechanism above is strategyproof, recall the results from Section 2.1, and specifically Corollary 2.4 which shows that an item-size monotone mechanism is strategyproof.

**Theorem 5.4** (Strategyproofness). *The Dual-Predictor mechanism is item-size monotone and hence strategyproof.*

*Proof.* Let the true processing times be given by $\boldsymbol{p}$, and consider any other report $\boldsymbol{p}_i$ made by machine $i$. If $\boldsymbol{x} = \boldsymbol{x}(\boldsymbol{p}_i, \boldsymbol{p}_{-i})$ and $\boldsymbol{x}' = \boldsymbol{x}(\boldsymbol{p}'_i, \boldsymbol{p}_{-i})$ are the allocations under the correct reports and the misreport respectively, we want to show that

$$(x_{ij} - x'_{ij})(p_{ij} - p'_{ij}) \leq 0. \tag{10}$$

Let $\mathsf{small}(j)$ and $\mathsf{small}'(j)$ denote the sets corresponding to the runs of the mechanism when given reports $(\boldsymbol{p}_i, \boldsymbol{p}_{-i})$ and $(\boldsymbol{p}'_i, \boldsymbol{p}_{-i})$ respectively.

1. **Case I**: Suppose $x_{ij} = 1$, i.e., the job $j$ is assigned to machine $i$ under the correct reports. Since $x'_{ij} \in \{0, 1\}$, (10) follows immediately in case $p_{ij} \leq p'_{ij}$, so it suffices to show that $p'_{ij} < p_{ij} \implies x'_{ij} = 1$. We consider two cases based on the size of $\mathsf{small}(j)$.

Suppose $\mathsf{small}(j) = \varnothing$, then for machine $i$ to be assigned job $j$, it must be that $i = \arg\min_{i'} p_{i',j}$. Since $p'_{ij} \leq \widehat{T}$, it must be the case that $|\mathsf{small}'(j)| \leq 1$. If $\mathsf{small}'(j) = \varnothing$, then $p'_{ij} < p_{ij}$ means we still have $i = \arg\min_{i'} p'_{i',j}$. Else $\mathsf{small}'(j) = \{i\}$, and then the mechanism always allocates job $j$ to the unique machine $i$ in $\mathsf{small}'(j)$. This means $x'_{ij} = 1$, proving the claim for the case when $\mathsf{small}(j) = \varnothing$.

Else $|\mathsf{small}(j)| \geq 1$. Since $j$ is assigned to machine $i$, we infer that $i \in \mathsf{small}(j)$ and $\widehat{\beta}_i \, p_{ij} \leq \widehat{\beta}_{i'} \, p_{i',j}$ for all $i' \in \mathsf{small}(j)$. (This is vacuously true when $|\mathsf{small}(j)| = 1$.) Since $p'_{ij} < p_{ij}$ and the other sizes $p_{i',j}$ remain unchanged for $i' \neq i$, we get $\mathsf{small}'(j) = \mathsf{small}(j)$. Moreover, since $\widehat{\beta}_i \geq 0$, the job $j$ will continue to be assigned to machine $i$, and hence $x'_{ij} = 1$.

2. **Case II**: Suppose $x_{ij} = 0$. Since $x'_{ij} \in \{0, 1\}$, (10) follows immediately in case $p'_{ij} \leq p_{ij}$, so it suffices to show that $p'_{ij} > p_{ij} \implies x'_{ij} = 0$.

Let $i_j$ be the machine that is assigned the job $j$ in the assignment $x$: i.e., $x_{i_j,j} = 1$. If $\mathsf{small}(j) = \varnothing$, then $p_{ij} > p_{i_j j}$ since it is not assigned the job $j$. Now, $p'_{ij}$ is even higher, and hence $x'_{ij} = 0$.

Else, $\mathsf{small}(j) \neq \varnothing$, which means that $i_j \in \mathsf{small}(j)$ with $\widehat{\beta}_{i_j}\,p_{i_j j} \leq \widehat{\beta}_{i'}\,p_{i'j}$ for all machines $i' \in \mathsf{small}(j)$. In case $p'_{ij} > \widehat{T}$, we have that $p'_{ij} \notin \mathsf{small}'(j)$ and therefore $x'_{ij} = 0$. Else if $p_{ij} < p'_{ij} \leq \widehat{T}$, then we have that

$$\widehat{\beta}_{i_j}\,p_{i_j j} \leq \widehat{\beta}_i\,p_{ij} < \widehat{\beta}_i\,p'_{ij}$$

and therefore, $x'_{ij} = 0$.

This case analysis implies that any machine $i$ misreporting its processing time for job $j$ leads to an allocation satisfying the item-wise monotonicity property (10). Now Corollary 2.4 implies that the mechanism is strategyproof. $\qquad\square$

## 6  Closing Remarks

This work extends the investigation of learning-augmented algorithms for optimization problems for which the pessimistic lower bounds can be circumvented using machine-learned predictions. We considered the unrelated machine scheduling problem in the strategic settings, where the agents (machines) can lie about the sizes of jobs to get an allocation with lower load. We show how to use only a linear number of predictions (essentially one value for each job and for each machine) to get constant consistency and optimal $O(n)$ robustness, improving on previous work that either achieved suboptimal robustness with linearly many predictions, or used quadratically many predictions for the optimal results. Our approach used a new LP relaxation with small integrality gap that also controlled the range of the dual values.

Several tantalizing directions remain open: can we improve the bounds even further? Since a typical case is when the number of machines is much smaller than the number of jobs, can we use only $O(n)$ predictions and get similar algorithmic guarantees? Can we predict these predictions much faster than predicting job sizes? Finally, what other algorithmic problems can use this "range-controlled dual" idea to get improvements?

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

## A  Learnability of Predictions

Let us show how we can learn predictions of the processing time, say in a PAC setting, where each instance $\boldsymbol{p}$ is obtained by first drawing a context/feature $a$ from a distribution $\mathcal{D}$ over some set $A$, and getting the matrix $n \times m$ of processing times using an unknown map $f(a) \in P := \mathbb{R}_+^{n \times m}$. Our goal is to learn some map $\widehat{f} : A \to P$ such that

$$\Pr_{a \sim \mathcal{D}} \left[ \forall i, j, \ \left| \log \widehat{f}_{ij}(a) - \log f_{ij}(a) \right| \leq \varepsilon \right] \geq 1 - \delta.$$

Then, at the time the mechanism is run on the true input, we suppose that the predictor sees the actual contexts/features $a$, and gives the prediction $\widehat{f}(a)$ to the algorithm. The guarantee above ensures that $\Pr \left[ D(f(a), \widehat{f}(a)) \geq e^\varepsilon \right] \leq \delta$, where the error parameter $e^\varepsilon \approx (1 + \varepsilon)$ for small $\varepsilon$.

Let us sketch one way to solve this learning problem. First, we can focus on each coordinate $f_{ij}$ separately and learn it with confidence $1 - \delta' := 1 - \delta/(mn)$; a trivial union bound then gives the claim. The goal now becomes that of learning a real-valued univariate function $g : A \to \mathbb{R}_+$, where $g(a) = \ln f_{ij}(a)$, to an additive accuracy of $\varepsilon$ with high confidence $1 - \delta'$. At this point, we can use standard tools from machine learning theory to give bounds on the number of samples that suffice to learn $g$ to an additive error of $\varepsilon$. Of course, one needs to make some assumptions on the complexity of the map $\tilde{g}$, which would depend on the applications at hand.

As a concrete example of these results, let us assume that the processing times lie in the range $[1, \ldots, e^K]$, and define $\tilde{g}(a) := \lfloor g(a) \rfloor$, then the function $\tilde{g}$ becomes $\{0, 1, \ldots, K\}$-valued. Furthermore, the gap due to this approximation $|\tilde{g} - g| \leq 1$; translating back via the definitions, the true processing time $f_{ij}(a)$ and its estimate $\hat{f}_{ij}(a) = \exp(\tilde{g}(a))$, would differ by at most a constant factor. Finally, given this $\{0, 1, \ldots, K\}$-valued function $\tilde{g}$, we can use the results of Ben-David et al. (1995) on learning such functions exactly with high confidence; this work reduces the learning of this function to showing the finiteness of VC-dimension of some related binary classification problems. In all these cases, the sample complexity behaves like $O((d_\Psi/\varepsilon^2) \log 1/\delta)$, where the parameter $d_\Psi$ is some notion of the dimension of associated concept classes.

## B  Error tolerant Dual-Predictor Mechanism

It can be shown that the DUAL-PREDICTOR mechanism is somewhat brittle — even in the presence of minor errors, the mechanism achieves the robustness guarantee. In this section, we design an error tolerant mechanism that builds on the previous mechanism and achieves a constant approximation factor not only when the predictions are accurate, but also when the predictions are approximately accurate.

Specifically, our strategyproof mechanism ERROR-TOLERANT DUAL-PREDICTOR will take as input an error threshold $\eta \geq 1$ in addition to the predictions used in Mechanism 1: it will output an allocation with consistency guarantees if the error of the prediction is at most $\eta$, and it will always ensures the robustness guarantees. (Both the guarantees now depend on $\eta$.)

### B.1  Approximately Correct Predictions

To begin, define the following "distortion" between two vectors $\boldsymbol{p}, \boldsymbol{q}$ of processing times:

$$D(\boldsymbol{p}, \boldsymbol{q}) := \max_{i,j} \max \left\{ \frac{p_{ij}}{q_{ij}}, \frac{q_{ij}}{p_{ij}} \right\}. \tag{11}$$

Observe that distortion is symmetric, so that $D(\boldsymbol{p}, \boldsymbol{q}) = D(\boldsymbol{q}, \boldsymbol{p})$.

**Definition B.1** (($\gamma, \eta$)-Approximate Correctness)**.** We now say that predictions $(\widehat{T}, \widehat{\beta}, \widehat{\varphi})$ are $(\gamma, \eta)$-*approximately correct* with respect to the (true) processing times $\boldsymbol{p}$ if there exist

  (a) processing times $\boldsymbol{q}$ such that the distortion $D(\boldsymbol{q}, \boldsymbol{p}) \leq \eta$, and
  (b) predictions $(\eta \widehat{T}, \widehat{\beta}, \widehat{\varphi})$ are $\gamma$-correct for processing times $\boldsymbol{q}$, where $\gamma$-correctness is defined as in Definition 5.2.

We say that the processing times $\boldsymbol{q}$ and estimate $\widehat{T}$ *certify* $(\gamma, \eta)$-approximate correctness.

## B.2 Error-Tolerant Mechanism

The error-tolerant mechanism ERROR-TOLERANT DUAL-PREDICTOR is given a parameter $\eta \geq 1$, along with predictions that are claimed to be $(\eta, \gamma)$-approximately correct, and it proceeds very similarly to the original DUAL-PREDICTOR mechanism: building a set of "small" machines for each job and then using a weighted greedy mechanism to assign to one of them (or to the machine with smallest reported processing time) if the set is empty.

---

**Mechanism 2:** Error-Tolerant Dual-Predictor

---

**Input:** Reported values $p_{ij}$ for all $(i, j) \in M \times J$, predictions $\widehat{\boldsymbol{\beta}} \in \mathbb{R}^n_+$, map $\widehat{\varphi} : J \to M$, and $\widehat{T} \in \mathbb{R}_+$.

**Output:** Assignment $\boldsymbol{x} \in \{0, 1\}^{m \times n}$.

**for** *each job $j$* **do**

    Let $\mathsf{small}(j) := \{i \mid p_{ij} \leq \widehat{T}\}$.

    **if** $p_{\widehat{\varphi}(j),j} \leq \eta^2 \widehat{T}$ **then**

        $\lfloor$ $\mathsf{small}(j) \leftarrow \mathsf{small}(j) \cup \{\widehat{\varphi}(j)\}$

    **if** $\mathsf{small}(j) = \varnothing$ **then**

        $\mid$ $\varphi(j) \leftarrow \arg\min_i p_{ij}$.

    **else**

        **let** $\overline{\beta}_{\widehat{\varphi}(j)} \leftarrow \widehat{\beta}_{\widehat{\varphi}(j)}$ **and** $\overline{\beta}_i \leftarrow \eta^2 \widehat{\beta}_i$ for all $i \neq \widehat{\varphi}(j)$

        $\varphi(j) \leftarrow \arg\min\{\overline{\beta}_i \, p_{ij} \mid i \in \mathsf{small}(j)\}$.        // breaking ties in favor of $\widehat{\varphi}(j)$.

**return** the vector $\boldsymbol{x}$ corresponding to this allocation $\varphi$.

---

There are two differences between the DUAL-PREDICTOR mechanism and this error-tolerant version: first, we now add in the predicted machine $\widehat{\varphi}(j)$ to the set $\mathsf{small}(j)$ even if the reported size $p_{\widehat{\varphi}(j),j}$ is larger than the threshold $\widehat{T}$, as long as it is not too large. Secondly, we modify the multipliers $\widehat{\boldsymbol{\beta}}$ to give the predicted machine an advantage, before choosing the machine for job $j$. Observe that for the setting of $\eta = 1$, the two mechanisms coincide.

## B.3 Robustness

The proof of robustness is quite similar to that of the original mechanism.

**Theorem B.2.** *Given any predictions, $\mathsf{MS}(\boldsymbol{p}, \boldsymbol{x}) \leq (1 + c)n \cdot \eta^2 \cdot \mathsf{MS}(\boldsymbol{p}, \boldsymbol{x}^\star)$.*

*Proof.* Let $\varphi$ and $\varphi^\star$ be maps from $J$ to $M$ that correspond to the assignments $\boldsymbol{x}$ and $\boldsymbol{x}^\star$ in the natural way. Define

$$r_j := \max(1, cn \cdot \widehat{\beta}_{\varphi^\star(j)}) \cdot \eta^2. \tag{12}$$

To begin, we claim that for any job $j$,

$$p_{\varphi(j),j} \leq r_j \cdot p_{\varphi^\star(j),j}. \tag{13}$$

Indeed, consider the various cases:

1. If $\mathsf{small}(j) = \varnothing$, then $p_{\varphi(j),j} \leq p_{i,j}$ for all machines $i$, and hence also for machine $\varphi^\star(j)$.

2. Else if $|\mathsf{small}(j)| \geq 1$, we consider two subcases.

(a) If $\varphi^\star(j) \notin \mathsf{small}(j)$, then $\widehat{T} < p_{\varphi^\star(j),j}$. Moreover, we assign job $j$ to some job in $\mathsf{small}(j)$, and therefore $p_{\varphi(j),j} \leq \eta^2 \widehat{T}$. Chaining the two implies $p_{\varphi(j),j} < \eta^2 \cdot p_{\varphi^\star(j),j}$.

(b) Else both $\varphi(j)$ and $\varphi^\star(j)$ belong to $\mathsf{small}(j)$. The mechanism's selection criterion ensures that

$$\overline{\beta}_{\varphi(j)} \, p_{\varphi(j),j} \leq \overline{\beta}_{\varphi^\star(j)} \, p_{\varphi^\star(j),j} \implies \widehat{\beta}_{\varphi(j)} \, p_{\varphi(j),j} \leq \eta^2 \widehat{\beta}_{\varphi^\star(j)} \, p_{\varphi^\star(j),j}.$$

Since the $\widehat{\beta}$ values range between $1/cn$ and 1, the claim immediately follows.

This proves the claim (13). Finally a calculation similar to the proof from Theorem 5.1 shows that the load of machine $i$ is at most $\eta^2 \cdot (n + cn) \cdot \mathsf{MS}(\boldsymbol{p}, \boldsymbol{x}^\star)$, completing the proof. $\qquad\square$

## B.4 Consistency

Now suppose that the predictions $(\widehat{T}, \widehat{\beta}, \widehat{\varphi})$ are $(\gamma, \eta)$-approximately correct, and let $\boldsymbol{q}$ be the processing times that certify this $(\gamma, \eta)$-approximate correctness. (Note that $\boldsymbol{q}$ is not known to the mechanism.) Unwrapping the definition, we get that

1. $D(\boldsymbol{p}, \boldsymbol{q}) \leq \eta$, so that $\boldsymbol{p}/\eta \leq \boldsymbol{q} \leq \eta\boldsymbol{p}$,
2. $\mathsf{OPT}(\boldsymbol{q}) \leq \eta\widehat{T} \leq \gamma\,\mathsf{OPT}(\boldsymbol{q})$,
3. $\widehat{\boldsymbol{\beta}} = \widetilde{\boldsymbol{\beta}}$ corresponds to an optimal dual solution $(\widetilde{\boldsymbol{\alpha}}, \widetilde{\boldsymbol{\beta}})$ to the dual $D_c(\eta\widehat{T}, \boldsymbol{q})$, and
4. $\widehat{\varphi}(j)$ corresponds to the solution $\boldsymbol{x}^\dagger$ obtained by rounding an optimal fractional solution $\widetilde{\boldsymbol{x}}$ to the primal $P_c(\eta\widehat{T}, \boldsymbol{q})$.

**Lemma B.3** (Consistency). *If the predictions are $(\gamma, \eta)$-approximately correct, then*

$$MS(\varphi, \boldsymbol{p}) \leq \widehat{\eta}^2 \cdot (2 + 1/(c-1)) \cdot \widehat{T} \leq \gamma\widehat{\eta}^2 \cdot (2 + 1/(c-1)) \cdot \mathsf{OPT}(\boldsymbol{p}),$$

*where $D(\boldsymbol{p}, \boldsymbol{q}) = \widehat{\eta} \leq \eta$.*

*Proof.* We first argue that the set $\mathsf{small}(j)$ is non-empty. Indeed, the machine $\widehat{\varphi}(j)$ lies in the support of some primal solution to the linear program $P_c(\eta\widehat{T}, \boldsymbol{q})$ and hence $q_{\widehat{\varphi}(j), j} \leq \eta\widehat{T}$. Since $p_{i,j} \leq \eta\,q_{i,j}$ for all $(i, j)$, we have $p_{\widehat{\varphi}(j), j} \leq \eta^2\widehat{T}$. By the definition of the mechanism, the machine $\widehat{\varphi}(j)$ is added to the set $\mathsf{small}(j)$ if it satisfies this last inequality, thereby ensuring that $\mathsf{small}(j)$ is non-empty. Hence, the mechanism assigns to the machine in $\mathsf{small}(j)$ minimizing $\overline{\beta}_i\,p_{ij}$.

Now consider the machines in $\mathsf{small}(j) = \{i \mid p_{ij} \leq \widehat{T}\} \cup \{\widehat{\varphi}(j)\}$. Since $\boldsymbol{p} \geq \widehat{\boldsymbol{p}}/\eta$, the former set is contained in $\{i \mid q_{ij} \leq \eta\widehat{T}\}$. Morever, as argued in the previous paragraph, $\widehat{\varphi}(j)$ also belongs to this set. In other words, the pairs $\{(i, j) \mid i \in \mathsf{small}(j)\}$ are contained in the permissible edges $E(\eta\widehat{T}, \boldsymbol{q})$.

Next, the feasibility for the dual program $D_c(\eta\widehat{T}, \boldsymbol{q})$ means that $\widetilde{\alpha}_j \leq \widetilde{\beta}_i\,q_{ij}$ for all $(i, j) \in E(\eta\widehat{T}, \boldsymbol{q})$; moreover, complementary slackness implies that $\widetilde{\alpha}_j = \widetilde{\beta}_i\,q_{ij}$ for machines $i \in \mathsf{small}(j)$ with $\widetilde{x}_{ij} > 0$, i.e., in the support of the optimal fractional solution. Hence, the machine $\widehat{\varphi}(j)$ is a minimizer of $\widetilde{\beta}_i q_{i,j} = \widehat{\beta}_i q_{i,j}$ over machines $i \in \mathsf{small}(j)$. However, we do not have access to the processing times $q_{ij}$ but only to $p_{ij}$. To fix this issue, we use (i) the fact that $q_{ij}/\eta \leq p_{ij} \leq \eta\,p_{ij}$ and (ii) the definition of $\overline{\beta}$ (as in in the mechanism) to infer that $\widehat{\varphi}(j)$ is the minimizer of $\overline{\beta}_i p_{ij}$ over machines $i \in \mathsf{small}(j)$. This means $\boldsymbol{x} = \widehat{\boldsymbol{x}} = \boldsymbol{x}^\dagger$.

Now suppose that $D(\boldsymbol{p}, \boldsymbol{q}) = \widehat{\eta} \leq \eta$. Then the resulting makespan is

$$\max_i \sum_{j:\varphi(j)=i} p_{ij} \leq \widehat{\eta} \cdot \max_i \sum_{j:\widehat{\varphi}(j)=i} q_{ij} = \widehat{\eta} \cdot MS(\widehat{\varphi}, \boldsymbol{q})$$

$$\leq \widehat{\eta} \cdot (2 + 1/c-1) \cdot \eta\widehat{T}$$
$$\leq \widehat{\eta} \cdot (2 + 1/c-1) \cdot \gamma\,\mathsf{OPT}(\boldsymbol{q})$$
$$\leq \widehat{\eta}^2 \cdot (2 + 1/c-1) \cdot \gamma\,\mathsf{OPT}(\boldsymbol{p}),$$

which completes the proof. □

## B.5 Strategyproofness

**Theorem B.4** (Strategyproofness). *The error-tolerant Dual-Predictor mechanism is strategyproof.*

The proof is essentially identical to the proof of Theorem 5.4, so we only sketch it here.

*Proof.* (Sketch) For a job $j$, if a machine $i$ was not in $\mathsf{small}(j)$, the only way for machine $i$ to become part of $\mathsf{small}'(j)$ is to reduce its processing time for the job $j$. Likewise, the only way to be removed from $\mathsf{small}(j)$ is to increase its processing time. Therefore, if machine $i$ already had the job $j$ ($x_{ij} = 1$), reducing its processing time ensures that it still receives the job, irrespective of whether $\mathsf{small}(j)$ is empty or not.

On the other hand, if machine $i$ did not receive the job, the case when $\mathsf{small}(j) = \varnothing$ is trivial. If $\mathsf{small}(j) \neq \varnothing$, increasing its processing time can only increase its scaled processing time until it eventually does not belong to $\mathsf{small}'(j)$; this shows $x_{ij} = x'_{ij} = 0$ as required. □

