# OpenReview forum: "Parsimonious Predictions for Strategyproof Scheduling"
_NeurIPS.cc/2025/Conference — NeurIPS 2025 poster_

### Official Review · Reviewer_4V1n · 2025-06-23

**Clarity:** 4
**Significance:** 3
**Originality:** 3
**Rating:** 5
**Confidence:** 4

**Summary:**

This paper addresses the problem of scheduling m jobs on n strategic, unrelated machines to minimize makespan, a setting where worst-case analysis shows that any strategyproof mechanism has a pessimistic guarantee. To bypass this limitation, the authors use the learning-augmented framework, designing a mechanism that uses parsimonious (i.e., simple) predictions. The core of their method is an LP relaxation designed to control the range of its dual variables, which are then used as predicted machine "biases”.  The proposed mechanism takes as input a predicted optimal makespan, a predicted bias for each machine, and a predicted "good" machine for each job. It achieves a makespan that is (1+c)n times the optimum regardless of prediction quality (robustness) and (2+1/(c−1)) times the optimum when predictions are correct (consistency), essentially matching the performance of previous work that required complex predictions.

**Questions:**

For NeurIPS, I think it would be valuable to give a little bit of practical context about the unrelated machine scheduling problem in strategic settings — i.e., what are some applications that this problem models (or approximately models)?

Minor comment: on line 180, I believe “assims” should be “assigns”.

**Ethical Concerns:**

["NO or VERY MINOR ethics concerns only"]

**Final Justification:**

I find the work interesting and believe it is a strong contribution to the field of learning-augmented algorithms and NeurIPS as a whole.  I appreciate the authors' efforts to contextualize some of the practical applications of the problem and hope to see some brief discussion of this in the final version.  I maintain my initial positive score.

**Limitations:**

Yes.

**Quality:**

3

**Strengths And Weaknesses:**

**Strengths**
A very well-written paper on a nice problem.  Analysis appears to all be sound, and the results are surprising in that parsimonious (i.e., succinct/simple) predictions can achieve the same (nearly optimal) consistency/robustness trade off for this problem.  The authors achieve this result via a new linear programming relaxation and dualization for the problem that may be of independent interest.

**Weaknesses**
My sense is that learning-augmented papers at NeurIPS typically include some small numerical studies to show how the newly proposed algorithms compare against baselines — such a study would be a valuable addition to this paper.  I would guess that those results would strictly increase the paper’s strength, since algorithms that use parsimonious predictions often compare quite favorably against those that use more complicated predictions models.

---

> ### Author Rebuttal · Authors · 2025-07-30
>
> Thanks for your encouragement and comments: we are happy that you found our results interesting and surprising (we did too!) and liked the presentation. We address your comments below:
>
> 1. **Numerical Studies**: This is a great suggestion, thanks: while our focus was on the theoretical aspects, looking at the empirical performance on natural classes of instances would be a nice direction for future work.
>
> 2. **Practical context**: Resource assignment problems are already hard to optimize, even in centralized systems. A common example is allocating jobs across cloud compute providers, where tasks need to be assigned to machines with different computational resources. When we move to more open and decentralized systems like those on the Internet, the problem becomes harder. Here, the resources belong to different agents who may act in their own interest and may not cooperate unless they are given the right incentives.
>
> In such settings, it is often important to ensure notions of non-manipulability and fairness. This could be for reasons related to social welfare or because of legislative requirements. Strategyproof mechanisms are useful in these situations because they make sure that no single powerful agent can manipulate the outcome to their advantage.
>
> Scheduling problems in practice can take many forms where fairness matters more than revenue. For example, in healthcare, different departments often manage their own resources. They might misreport their capacity to reduce their workload or to increase their revenue. Without a strategyproof mechanism, this can lead to poor scheduling, delays, and less access for patients.
>
> There are also other examples like peer reviews or job assignments through bidding, especially when play money is involved. These systems can also be manipulated and they do not have the dimension of revenue maximization. While they may not exactly match the unrelated machine scheduling model, they still show the same basic need: to design mechanisms that work well even when agents behave strategically.
>
> We will add some applications in the next version.
>
> 3. **Comment on line 180**: We will fix this, thanks.

---

> > ### Comment · Reviewer_4V1n · 2025-08-05
> >
> > Thank you for your response. My questions are well-addressed, and I will take this response into account for my final recommendation.

---

### Official Review · Reviewer_gcXb · 2025-06-27

**Clarity:** 3
**Significance:** 3
**Originality:** 3
**Rating:** 5
**Confidence:** 3

**Summary:**

The paper considers learning-augmented truthful mechanisms for makespan scheduling on unrelated machines. In makespan scheduling on unrelated machines, we are given $m$ jobs to be scheduled on $n$ machines, and the goal is to assign the jobs to the machines with the objective to minimize the maximum processing load assigned to a single machine. The processing time $p_{ij}$ of a job $j$ depends on the machine $i$ to which it is assigned. In this paper, the authors consider a strategic variant of this problems, where the machines are selfish agents that might misreport their processing times for the jobs in order to receive a smaller processing load. The goal is to design a strategyproof scheduling mechanism in which no agent has an incentive to misreport their processing times, while still minimizing the objective of the underlying scheduling problem. In this setting, the best-possible approximation ratio of any strategyproof mechanism is $n$. In the learning-augmented setting, the mechanism has access to untrusted predictions on the problem instance. If the meachnism has access to predicted processing times for all job-machine-pairs, then there is a $\mathcal{O}(1)$-consistent and $\mathcal{O}(n)$ -robust algorithm (Balkanski et al. (2023)), where the consistency is the approximation ratio for perfect predictions and the robustness is the approximation ratio for arbitrary predictions. If the number of predicted values is linear, then the previously best-known learning-augmented algorithm is $\mathcal{O}(1)$-consistent and $\mathcal{O}(n^2)$-robust (Christodoulou et al. 2024). The main contribution of this paper is a $\mathcal{O}(1)$-consistent and $\mathcal{O}(n)$-robust algorithm using a linear number of predicted values.

To achieve this result, the authors assume access to a predicted optimal job assignment, a predicted optimal makespan, and predicted optimal variables for the dual of a certain LP-relaxation. At its core, the presented mechanism greedily matches a job $j$ to the machine $i$ of minimum $p_{ij}\beta_i$, where $p_{ij}$ is the reported processing time of job $j$ on machine $i$ and $\beta_i$ is the predicted dual variable, provided that the reported time is smaller then the prediceted optimal makespan. The predicted assignment is only used for tiebreaking. The key idea of the analysis is the usage of a new LP-relaxation that guarantees the dual variables to be within a certain range. Assuming access to predictions on these dual variables, the authors can now use that the predicted values are within this range, which is crucial for proving the improved guarantees. In the appendix, the authors give an error-tolerant version of this mechanism.

**Questions:**

* How does the error-tolerant algorithm behave if the predicted $\beta$-values are slightly infeasible, i.e., outside the range $[1/cn,1]$? Does it immediately fall back into the robustness case, or does it still achieve improved guarantees?
* Apart from the new LP-relaxation, what would you consider the main technical contribution of your paper?

**Ethical Concerns:**

["NO or VERY MINOR ethics concerns only"]

**Final Justification:**

In my opinion, the paper is a very strong contribution. As outlined in my original review, the results of the paper essentially match the previous best-known tradeoff while using smaller predictions and simplifying the algorithm and analysis. The new LP-relaxation is a strong and elegant technical contribution. My only real concern regarding the brittleness w.r.t the $\beta$-values was adequately addressed  in the author's rebuttal. Overall, I maintain my initial positive estimation of the paper and recommend to accept it.

**Limitations:**

yes

**Quality:**

3

**Strengths And Weaknesses:**

Strength:
* The presented algorithm essentially matches the previously best-known consistency and robustness trade-off, while using significantly smaller predictions ($\mathcal{O}(n+m)$ instead of $\mathcal{O}(n \cdot m)$ ), and significantly improves upon the previously best-known guarantees for predictions of a linear size. This seems to be significant progress towards the best-possible trade-off for small predictions.
* The main technical contribution to achieve this result is a new LP-relaxation. This new LP-relaxation allows to bound the dual variables within a certain range, which is crucial for the proof of the consistency and robustness guarantees. In my opinion, this is a new and very elegant idea to achieve improved results.
* Since many learning-augmented algorithms in the literature are based on predicted dual variables, the used techniques could potentially also be applicable in different contexts, which raises intriguing follow-up questions.

Weaknesses:
* In the appendix, the authors show how to design an error-tolerant version of their mechanism, which is also less brittle (more robust against small perturbations in the predicted values). I agree that this approach gives an error-dependency and addresses the brittleness with respect to the predicted makespan and the predicted assignment. However, I am not sure about error-tolerance with respect to the predicted $\beta$-values. The original algorithm is defined to immediately discard infeasible predicted $\beta$-values, even if they are only outside the feasibility range by an $\epsilon$. The error-tolerant version seems to do the same (at least the proof of Theorem C.2 still assumes that the predicted values are feasible), which would mean that error-tolerant mechanism remains very brittle against slightly infeasible $\beta$-values.
* While the presented approach uses smaller predictions than the algorithm by Balkanski et al. (2023), it is completely unclear to me what the benefit of these smaller predictions is. As stated by the authors, it remains unclear whether it is easier to learn these smaller predictions than the $m \cdot n$ processing times. Since the presented learnability result is with respect to the processing times, it remains unknown whether it is possible to learn the smaller predictions without also learning the processing times. Even the error-threshold $\eta$ is with respect to the distortion of processing times and not a function of the actual predictions.
* Apart from the new LP-relaxation, most of the used techniques seem to be already established in the area of learning-augmented algorithm design, which somewhat limits the technical contribution beyond the new relaxation. Admittedly, you could also see this as a strength of the new relaxation rather than a weakness.

---

> ### Author Rebuttal · Authors · 2025-07-30
>
> Thanks for your encouragement and comments: we appreciate that you found the work interesting and elegant, and we agree that the ideas here may be useful broadly. Below, please find responses to your specific comments.
>
> **$\beta$ values slightly infeasible**: This is a nice question! Indeed, the algorithm should be robust to infeasibilities in the beta-values: if the beta-values are slightly infeasible, we can first scale them down to make them sum to 1. And then this would make them lie in $[1/(c’n), 1]$ for some slightly smaller c’. So the quality would suffer smoothly in how infeasible these beta values are. We will add a discussion about this in the next version.
>
> **fewer predictions are good**: For the second comment, we view having fewer predictions as giving us the power of compression: indeed, we can compute these predictions from predictions of the processing times, so they are at least as powerful. Moreover, these are fewer objects to store and process during the algorithm’s execution. (Indeed, we feel that our algorithm is conceptually cleaner as a result of these predictions.) Moreover, it is also a proof of concept, that a small number of natural quantities suffice (machine weights, and job-to-machine assignments). We hope that future work will be able to show how to learn these faster.
>
> **Additional technical contributions**: The other technical part would be the simple algorithm, which goes hand-in-hand with the LP. The previous algorithm of Balkanski et al. was not very intuitive, and more difficult to interpret, whereas we feel our algorithm follows naturally from the LP, and leads into the analysis quite naturally as well. We do view the simplification of the algorithm (and, as you say, the use of classical techniques, as a technical benefit, since we feel it exposes more of the essential ideas of the problem and its solution. Besides these technical benefits, our contributions are more conceptual: how we can make the predictions more parsimonious using LPs in non-trivial ways, and how new LPs may need to be devised to work well for problems related to learning augmented algorithms.

---

> > ### Comment · Reviewer_gcXb · 2025-08-03
> >
> > Thank you to the authors for their helpful response, especially for the explanation regarding the β-values. This resolves my (very minor) concerns. I will keep my positive estimation of the paper.

---

### Official Review · Reviewer_x1Zx · 2025-07-01

**Clarity:** 4
**Significance:** 2
**Originality:** 2
**Rating:** 3
**Confidence:** 3

**Summary:**

The paper considers the unrelated machine scheduling to minimize the makespan. The processing time matrix is not known a priori. The machines are strategic agents, meaning that they may misreport processing times to benefit themselves, i.e., lower their processing load. The problem is to design a strategyproof mechanism where misreporting can only lower each agent’s utility and minimize the makespan. The problem is considered under the algorithms with predictions setting, where the mechanism is given predictions about makespan, machine loads, and the job-to-machine mapping. The problem of strategyproof scheduling has been studied where the predicted processing times are accessible and a constant consistent and O(n) robust algorithm exists. The author proposes a strategyproof mechanism based on linear program relaxation and rounding, with the number of predictions required being linear in the number of machines and jobs. The consistency is constant and robustness is in O(n), i.e., the machine is approximately optimal if the predictions are correct and O(n) times the optimum under arbitrarily bad predictions.

**Questions:**

1. It seems that the predicted job-to-machine map phi is not as important, which is used just for tie-breaking. Is it possible to drop this via still achieving the same robustness and consistency?

2. Can the authors justify strong sensibility and rationale for the prediction error metric — gamma-correct predictions? If the authors can show that a gamma-correct prediction for some gamma is always (or nearly always) achievable via some learning techniques, the reviewer may consider a re-evaluation.

**Ethical Concerns:**

["NO or VERY MINOR ethics concerns only"]

**Quality:**

3

**Strengths And Weaknesses:**

Strengths:
1. The problem is well-presented.
2. The contribution of reducing the number of predictions is critical.
3. The key metrics of algorithms with predictions are analyzed, e.g., consistency and robustness.

Weaknesses:
Although the attempt to reduce the number of predictions is meaningful, the total contribution does not seem to pass the bar of NeurIPS. It did not at least convince the reviewer that it was the case. I will elaborate below.

The definition of prediction error is too tailored to the analysis and the algorithm and it is very strict. There are two weaknesses: 1) the prediction for the makespan is at least the optimum and 2) the predictions for machine load and assignment need to be exact so that the mechanism can output a solution (after rounding) that matches the optimal solution to the modified primal. First of all, it seems unclear how the bound of the makespan predictions can be achieved. Second, it is hard to examine the hardness of the second condition; it is unclear if the machine load and assignment can easily end up getting the exact outcome in practice. These two weaknesses make the error metric unintuitive and hard to examine its effectiveness in measuring the prediction quality. It seems possible that some set of predictions does not have correctness. It seems hard for any learning model to produce predictions that are gamma-correct for any gamma.

The weakness in the prediction error makes the consistency hard to interpret. It is still the case that the algorithm is constant consistent when all predictions are perfect, but it does not make meaningful sense when the predictions are not perfect and some predictions do not correspond to any gamma-correctness. Therefore, the reviewer has to assess the consistency not as a function of the prediction error, but as a constant for the case of perfect predictions. This drawback significantly lowers the technical contribution of the paper.

In my opinion, the assessment of prediction error definition is inherently discretionary. I assess it as indecent due to 1) being too strict and tailored to the algorithm and 2) non-coverage for all predictions. This makes the analysis for an error-dependent consistency less contributive — I have to interpret the consistency as being meaningful for the case of the perfect prediction but not the general case. This lowers the contribution of the paper to below the bar of acceptance.

Minor issues:
1. It is unclear when reading section 3 “a new linear program relaxation”, what the old is and what the contributions of the new one are. The authors include the old linear program in the appendix, but a clear discussion is needed for the readers to appreciate the contributions.
2. Incomplete sentence in Line 300, “complementary slackness implies that …”.

---

> ### Author Rebuttal · Authors · 2025-07-30
>
> Thanks for your comments and questions: we are happy to see you agree with the importance of parsimony in predictions, and  that you found the paper interesting and well-written.
>
> We respectfully disagree with some of the concerns you voice, and would like to suggest that the prediction error definition is quite natural under the circumstances. Let us elaborate:
>
> **Prediction error**:
> We feel that the predicted optimal makespan is a reasonable thing to predict for a collection of jobs. Most models output estimates with confidence intervals. If such a model predicts the makespan lies in some interval $[\mu_1, \mu_2]$ with high probability, we can output $\mu_2$ our estimate for the makespan. In this case the prediction would indeed be an overestimate for the makespan with high probability.
>
> As for the prediction of the optimal machine for jobs, we agree that there is no natural way to talk about “nearly-correct” assignments, since it is a mapping. As written, we want these predictions to conform to the rounding of the LP solution. But we can weaken the requirement considerably from what is given in the submission. It would be fine to have any predictions which ensure two properties: (a) ensure the assignment has low load, and (b) that the job is sent to some machine to which it has been sent fractionally. Indeed, previous work on the problem (at Neurips 2024) considered the optimal assignment as a prediction, but achieved a worse robustness/consistency tradeoff. We will add a discussion about this in the next version.
>
> The prediction error considered in our work is the same as that used in all earlier works, and it ties together all our different types of predictions under a single parameter, $\eta$. We could incorporate multiple error parameters, one per type of prediction, for example: predictions where the makespan is multiplicatively $\gamma$-correct, the assignment $\phi$ is $\eta$-correct (i.e., results in a makespan less than $\eta \cdot OPT$), and the $\beta$ values are multiplicatively $\varepsilon$-correct (baseline for correctness would be the beta values corresponding to the mapping $\phi$). An error-tolerant mechanism, similar to the one we describe in the appendix can be designed that still provides the same consistency and robustness, albeit with different constants. However, we chose to use a unifying error metric, as also used in earlier works.  This has the additional benefit of allowing us to compare our error-tolerant mechanism against the work of Balkanski et al., and we show that our constants match theirs. Highlighting that using fewer predictions does not lead to any deterioration, even in the constants.
>
> Lastly, in the learning-augmented algorithms model, it is generally assumed that the mechanism is equipped with predictions about the agents. We make no assumptions about how the predictions are produced and simply treat the predictions as additional information available to a plug-and-play algorithm that offers optionality.
>
> **Question on $\varphi$'s importance**: This is an interesting question. Indeed, we want to emphasize the importance of tie-breaking in our algorithm, and in the previous algorithms for this problem which give constant consistency. If we consider the non-strategyproof algorithms that give good “offline” approximations to makespan minimization, they round the LPs and make very coordinated decisions. If we want to do coordination in a strategyproof setting, this strong signal across machines has to come from the predictions. (Else we face the Omega(n) lower bound.) So we can use predictions of all the job sizes, but lacking that, we think these job-to-machine maps — or some surrogate for them — are probably necessary to get constant consistency.
>
> **On learnability of $\gamma$ correct predictions**: Apart from the discussion about the gamma-correct predictions above, we give a discussion of PAC-style learning techniques in Appendix A of the paper.
>
> Thank you also for pointing out the minor issues. We will make sure to fix them in the next version.

---

### Official Review · Reviewer_GfdH · 2025-07-02

**Clarity:** 3
**Significance:** 3
**Originality:** 4
**Rating:** 5
**Confidence:** 4

**Summary:**

This paper studies an important problem from the algorithmic game theory literature, strategy-proof scheduling. The authors show how it is possible to use a linear number of predictions (O(n+m) where m if the number of jobs and n is the number of machines) to improve on the approximations known when taking a worst-case analysis approach. They do so by using a novel framing of the problem, and match or improve on known results in the space, using less predictions/predicted information.

**Questions:**

1. I understood mathematically the role of the  predicted \beta’s in the dual formulation. Furthermore, the paper does state that they can be viewed as “machine weights” (line 183) or machine biases (line 245).  However, if I was to actually deploy the mechanism in practice, is there an intuitive way to obtain these weights/biases?

2. Theorem 5.1 provides a guarantee on the robustness of the mechanism, no matter what the prediction quality is. Theorem 5.2 looks at consistency and provides a more nuanced insight since the result depends on the quality of the prediction. Is it possible to revisit Theorem 5.1 and consider prediction quality? If so, would this require a different analytical approach?

3. Can you provide examples of other, specific domains where the proposed approach might be useful?

**Ethical Concerns:**

["NO or VERY MINOR ethics concerns only"]

**Final Justification:**

Thanks for the response. I continue to like the paper and my review doesn't change.

**Limitations:**

Yes

**Quality:**

3

**Strengths And Weaknesses:**

Overall this is a strong, technically interesting paper, that advances the state-of-the art in strategy-proof scheduling, a classic problem from algorithmic game theory.

Strengths:
- Studying a well-motivated problem (strategic scheduling) from the algorithmic game theory literature.
- Novel framing of the problem (i.e. a new framing of the underlying linear programming relaxation) which allows them to provide a nearly optimal tradeoffs with respect to consistency and robustness using a linear number of predictions.
- Appears to be technically sound and full proofs are provided in the appendix (with good proof sketches in the main paper).

Weaknesses:
- The paper is dense which limits its accessibility. While reading the work, I was trying to figure out how I might use the results in practice. While I understood the role of the beta's in the dual formulation, I was less confident when it come to understanding how I would "predict" them if I was considering them as features of the machine. More clarity here would have been useful.

---

> ### Author Rebuttal · Authors · 2025-07-30
>
> Thanks for your encouragement and comments. It's great to hear that you found the work strong and technically interesting. We address your comments below:
>
> **Paper is dense**: Please see below for a detailed discussion; we will elaborate more in the paper as well, and try to make the paper more accessible.
>
> **Way to obtain these weights/biases**: Note that in the setting of unrelated machines where jobs have very different processing times on different machines, the machine weights depend on features of the (job, machine) pairs, and not just the machines themselves. To get an intuitive understanding, it may be useful to consider the setting where the processing times of a job on some machine are of the form, say, $p_j * s_i$, where $p_j$ is a job-dependent term, and $s_i$ a machine-dependent term. In this case, one can hope for the machine weights to depend only on the machine features. One intuitive way to get these weights here would be to use a “multiplicative-weights” iterative approach, where we assign uniform weights, and then increase or reduce the weights based on whether the machine loads are too high or too low. Such a process would converge to near-optimal weights after a small number of rounds.
>
> **Theorem 5.1 and 5.2**: We would be hesitant to change Theorem 5.1, since the robustness property wants us to be good, regardless of the quality of the predictions. However, we do agree that one wants a smoother trade-off between the robustness and the consistency: we hope that the results of the Appendix (particularly Lemma C.3) give a partial answer to this, since they give guarantees even if the predictions are not correct, but not too incorrect.
>
> **Examples of other domains where these techniques can be useful**: This is something we are thinking about, where we are looking at other “load-balancing” problems in routing and scheduling, where we may be able to use a similar linear-programming approach. In general, packing and resource allocation are big areas where similar issues come up; there are more challenges there, but we hope to things interesting things to say.

---

> > ### Comment · Reviewer_GfdH · 2025-08-05
> >
> > Thanks for your response. I am curious as to how the general ideas could be applied to resource allocation problems -- it will be interesting to see how that plays out.
> > Anyway, I remain positive about the paper itself and the general direction it is taking.

---

### Decision · Program_Chairs · 2025-09-17

**Decision:**

Accept (poster)

**Comment:**

This paper studies strategyproof scheduling in the algorithms with predictions framework. The main result is a strategyproof mechanism for the classical problem of makespan minimization on unrelated machines that achieves O(1)-consistency (approximation when predictions are correct) and O(n)-robustness (approximation when predictions are incorrect) with O(m + n) predicted values. This result matches the consistency and robustness from previous work, but with significantly less predicted values (O(m+n) instead of mn).
The reviewers agreed that this is a strong result and that using a small number of predictions is crucial in algorithms with predictions. The technical contribution with the LP relaxation framework for this problem was also highly appreciated.